# In-hospital mortality outcomes of favipiravir in patients with moderate to severe COVID-19 infection: An emulated target trial using real-world data from the largest field hospital in Thailand

**Lalita Lumkul**[1,2‡], **Krittai Tanasombatkul**[1,3‡], **Phongsak Nitikaroon**[4],
**Thotsaporn Morasert**[5�*], **Phichayut Phinyo**[1,2,3�*]

1 Center for Clinical Epidemiology and Clinical Statistics, Faculty of Medicine, Chiang Mai University, Chiang Mai, Thailand, 2 Center of Multidisciplinary Technology for Advanced Medicine (CMUTEAM), Faculty of Medicine, Chiang Mai University, Chiang Mai, Thailand, 3 Department of Biomedical Informatics and Clinical Epidemiology (BioCE), Faculty of Medicine, Chiang Mai University, Chiang Mai, Thailand, 4 Office of the Permanent Secretary Ministry of Public Health, Nonthaburi, Thailand, 5 Pulmonary and Critical Care Medicine, Department of Medicine, Suratthani Hospital, Surat Thani, Thailand

œ These authors contributed equally to this work.
‡ These authors are co-first authors, LL and KT are also contributed equally to this work.
* thot_kwan@hotmail.com (TM); phichayutphinyo@gmail.com (PP)

## Abstract

### Background

Favipiravir, an antiviral agent, has been widely used to treat COVID-19 due to its potential mechanism of action, despite limited evidence of its efficacy in moderate to severe cases.

### Aim

This study aimed to evaluate the efficacy of favipiravir in improving in-hospital mortality outcomes among patients with moderate to severe COVID-19 through an emulation of a target trial.

### Methods

We emulated a target trial using observational data from Bussarakham field hospital, Thailand between May 14 and September 20, 2021. Patients were categorized into three groups: those receiving favipiravir with dexamethasone (FPV with Dexa), favipiravir alone (FPV), and symptomatic treatment (ST). In-hospital mortality within 30 days was the primary outcome.

### Results

From 18,184 patients admitted to the hospital, a total of 3,193 moderate to severe COVID-19 cases were included. Of these, 2,256 (70.65%) received FPV with Dexa,

**Data availability statement:** All relevant data are within the paper and its Supporting Information files.

**Funding:** The author(s) received no specific funding for this work.

**Competing interests:** The authors have declared that no competing interests exist.

828 (25.93%) received FPV, and 109 (3.41%) received ST. The restricted mean survival times were 29.68 days (95% CI: 29.52, 29.84) for FPV with Dexa, 29.46 days (95% CI: 29.22, 29.71) for FPV, and 28.14 days (95% CI: 26.51, 29.76) for ST. Only FPV showed marginally significant difference when compared to ST. However, there was a trend in prolonging survival time in FPV with Dexa group, and the results were more pronounced in severe and hypoxic patients.

## Conclusion

Our emulated target trial suggests favipiravir, especially with dexamethasone, offers a modest survival benefit in moderate to severe COVID-19, particularly in hypoxic patients. It supports favipiravir as a practical antiviral in settings where other antivirals are not available. Further randomized controlled studies are needed to confirm its role, alongside standard corticosteroid therapy.

## Introduction

Coronavirus disease 2019 (COVID-19) is a highly transmissible, severe acute respiratory illness caused by the SARS-CoV-2 virus presenting a significant challenge to global healthcare systems and prompting extensive investigations into several antiviral agents [1–2]. In Thailand, as in other countries, favipiravir was rapidly adopted as a first-line antiviral treatment for COVID-19 following its emergency use authorization in 2020 [3–6]. The drug was included in the Thai Ministry of Public Health (MoPH) guidelines and was widely used during the peak of the pandemic, particularly in response to the third wave driven by the Delta variant in 2021 [7]. This variant significantly increased the number of patients requiring hospitalization, and was associated with higher mortality and transmissibility [8,9]. The surge in cases led to the establishment of Thailand's largest national field hospital to manage patient overflow and ease the burden on healthcare facilities.

Despite its widespread use, favipiravir's clinical efficacy in COVID-19 remains controversial [2,10]. Several clinical trials focused primarily on mild COVID-19 cases, where favipiravir's effect is minimal, as most patients naturally recover with supportive care [11,12]. However, in moderate to severe patients who are facing higher risks of respiratory failure and complications, antiviral and anti-inflammatory therapies may play a crucial role in improving outcomes [13,14]. In addition, the study from RECOVERY trial demonstrated that dexamethasone, a potent anti-inflammatory steroid, reduced 28-day mortality in COVID-19 patients but not in those not receiving respiratory support [15]. Therefore, during the pandemic wave, the Thai MoPH guideline suggested the use of favipiravir in combination with dexamethasone in patients with moderate to severe symptoms.

In Thailand, studies investigating the use of favipiravir in COVID-19 patients are somewhat limited. Studies by Sirijatuphat et al. [16] and Siripongboonsitti et al. [17] compared efficacy of favipiravir with control treatments and found that favipiravir showed improvements in clinical outcomes in mild to moderate COVID-19 patients.

Rattanaumpawan et al. [18] reported favipiravir's effectiveness in reducing hospital stays among hospitalized patients, though its impact on mortality remained unclear. Furthermore, a real-world study during the Delta variant wave indicated that favipiravir monotherapy reduced the 28-day mortality risk in severe COVID-19 (Relative risk = 0.72; 95% CI: 0.58–0.91; P = 0.006) but not in mild and moderate patients [19]. The results are still inconclusive due to differences in trial protocols. Additionally, the causal effect of favipiravir in combination with dexamethasone in moderate to severe patients remains unexplored.

Therefore, this study aimed to evaluate the effectiveness of favipiravir in reducing in-hospital mortality among patients with moderate to severe symptoms admitted to a field hospital setting. We compared patients who received favipiravir monotherapy or favipiravir in combination with systemic dexamethasone to those who received symptomatic treatment as part of standard care. Our main objective was to provide causal evidence for the treatments through an emulation of a target trial using observational data from the largest field hospital in Thailand.

## Materials and methods

### Study design

This therapeutic research was conducted to determine the in-hospital mortality outcome of favipiravir with or without dexamethasone compared to standard symptomatic therapy in moderate to severe COVID-19 patients using electronic medical records of Bussarakham Field Hospital (BH), Thailand. Retrospective data were used to compare the survival outcomes between patients who received (i) favipiravir combined with dexamethasone (FPV with Dexa), (ii) favipiravir alone (FPV), and (iii) symptomatic treatment (ST). Since this study utilized retrospective data, the overall results may be subject to bias due to the absence of randomization and the presence of immortal time bias, which occurs when the start of treatment and the start of follow-up do not align. Therefore, the study was conducted based on an emulated target trial framework [20,21]. A summary of the hypothetical target trial protocol and the emulated trial, compared to original cohort, is provided in Table 1.

The Institutional Review Board and the Ethics Committee of Pranangklao Hospital approved the study protocol (Approval ID: EC38/2021). This study was based on retrospective data collection. All treatments followed the Ministry of Public Health (MoPH) management guidelines and were determined by the treating physicians, with no research-related interventions influencing clinical decisions. Because data were collected retrospectively from an existing database, obtaining informed consent from patients was not feasible and was waived due to the retrospective nature of data collection in a non-experimental setting. To ensure patient confidentiality and data protection, all collected data were fully anonymized excluding personally identifiable information such as hospital numbers, identification numbers, names, or dates of birth. De-identified patient data were accessed by the research team between 1st December 2021 and 28th November 2022.

### Study setting

The field hospital was operated by the MoPH and was located in the IMPACT Challenger Hall, the largest exhibition and conference hall of Thailand. The venue comprises of 3 connected halls, each of which was 20,000 square meters in size and was partitioned into 4 hospitalized areas: one for men, one for women, one for mixed-gender ward, and one for patients who need supplemental oxygen. There were more than 3,000 beds for patients with mild to severe conditions, 32 semi-Intensive Care Unit (ICU) beds, 17 ICU beds, 100 negative pressure rooms, as well as more than 100 life-support devices for patients who deteriorate rapidly to be transferred to a nearby hospital. During the study period, it was the third phase of the pandemic with the influence of Delta variant [7,22].

### Data collection

The study data were collected and managed using REDCap electronic data capture tools hosted at Yale University [23,24]. Patient's demographic, clinical investigation, treatment, and outcome data were extracted from the medical records of BH by a trained data collector. Age, body weight, body mass index (BMI), smoking status, comorbidities

**Table 1. Study components based on hypothetical target trial, emulated trial and original cohort.**

| Components | Target trial | Emulated cohort | Original cohort |
|---|---|---|---|
| **Design** | Randomized controlled, concurrent trial of COVID-19 patients with moderate to severe symptom | Emulated target trial from retrospective observational cohort of COVID-19 patients with moderate to severe symptom | Retrospective observational cohort of COVID-19 patients with moderate to severe symptom |
| **Aim** | Estimate the effect of favipiravir in combination with dexamethasone, or favipiravir alone on in-hospital mortality outcome. | Same as target trial | Same as target trial |
| **Eligibility** | COVID-19 patients with moderate to severe symptoms admitted to BH. | Same as target trial | COVID-19 patients with moderate to severe symptoms admitted to BH during May 14,2021 to September 20, 2021 |
| **Exclusion** | Patients who:<br>- had been infected > 7 days<br>- received *Andrographis paniculate*<br>- received mechanical ventilation<br>- received Favipiravir > 14 or ≤ 1 days<br>- had more than 1 day's delay in getting favipiravir | Same as target trial | Same as target trial |
| **Treatment strategies** | 1. Favipiravir with dexamethasone<br>2. Favipiravir<br>3. Symptomatic treatment | Same as target trial | Same as target trial |
| **Treatment assignment** | Patients were randomly assigned to each treatment strategy. | Trial emulation was conducted to generate clones for each patient and were allocated to all treatment groups. | Patients were classified to treatment groups based on the electronic medical record. |
| **Treatment implementation** | Treatments were administered immediately after randomization | | Treatments were given according to hospital's protocol. |
| **Outcome** | In-hospital mortality within 30 days of hospitalization. | Same as target trial | Same as target trial |
| **Follow up and censoring** | The follow-up started at the time of assignment to the treatment strategies until<br>- in-hospital mortality<br>- discharge (competing event)<br>− 30 days after admission (censoring) | Same as target trial | Same as target trial |
| **Adjustment variables** | Age, gender, underlying diseases (e.g., hypertension, obesity, COPD, cardiovascular disease), duration from PCR to admission, vital sign, vaccine status. | Same as target trial | Same as target trial |
| **Causal contrast** | Per-protocol analysis. | Per-protocol analysis since intended treatment could not be identified. Patients, and clones, were censored when they deviated from their assigned strategy at the time of deviation. | As-treated analysis according to the treatment received based on electronic medical record. |
| **Statistical analysis** | Differences in the restricted mean survival time among treatment group after admission. | Same as target trial | Same as target trial |

including hypertension, diabetes mellitus, cirrhosis, cerebrovascular accident (CVA), chronic kidney disease (CKD), chronic obstructive pulmonary disease (COPD), cardiovascular disease (CVD), and immunocompromised status were collected.

The clinical presentations — symptoms and admission parameters, laboratory investigations — complete blood count, C-reactive protein (CRP), lactate dehydrogenase (LDH), liver function tests (LFT), serum creatinine, and chest radiography, and management — antiviral therapy, steroid therapy, and oxygen support at the time of first admission were also extracted from BH medical records.

We retrieved vaccination data (type of vaccine and date of injection) using patients' national identification numbers. During the study period, three different vaccines were available in Thailand, including CoronaVac® (Sinovac, Beijing, China), Vaxzevria® (Oxford/AstraZeneca, Cambridge, United Kingdom) and Covilo® (Sinopharm, Beijing, China).

### Eligibility for the analysis set

The study domain included symptomatic patients aged ≥18 years who were diagnosed with COVID-19 and admitted to the field hospital from May 14 to September 20, 2021. All COVID-19 diagnoses were confirmed by reverse transcription polymerase chain reaction (RT-PCR) before admission.

Patients with risk factors for severe COVID-19 include those who aged ≥60 years old, or obesity (BMI > 35 kg/m² or body weight ≥ 90 kg), or those with at least one comorbidity. Disease severity was categorized according to the Thai MoPH guidelines [4,25] and WHO clinical management: a mild case refers to symptomatic patients without evidence of viral pneumonia or hypoxia; a moderate case refers to symptomatic patients with at least one risk factor and the need for respiratory support, including those with COVID-19 pneumonia disease but has an oxygen saturation >96%; a moderate-to-severe case is symptomatic patients with pneumonia and oxygen saturation ≤96%. This moderate-to-severe category is not defined in the WHO guideline and is generally categorized as moderate in the WHO classification; however, we specifically defined this group based on our MoPH which can better reflect the clinical management protocol during the pandemic [4,25].

To specifically evaluate the effectiveness of favipiravir and dexamethasone in moderate to severe COVID-19 cases in the field hospital, we included patients who were indicated for both favipiravir and dexamethasone, which is moderate case and moderate-to-severe cases otherwise, they were not included in the trial. In addition, we excluded patients receiving *Andrographis paniculate* extract, requiring mechanical ventilation, and who had been infected with COVID-19 for more than 7 days. Furthermore, we excluded patients who received favipiravir for less than 1 day or more than 14 days, or who started favipiravir more than 1 day after admission. Patients who received other medications, such as prednisolone, methylprednisolone, enoxaparin, and remdesivir, were also excluded.

### Treatment strategies and assignments

The MoPH guidelines and physician orders at Bussarakham Field Hospital (S1 Fig.) recommended a favipiravir dose of 1800 mg twice daily (BID) on day 1 and 800 mg BID on subsequent days (days 5–14) for COVID-19, and dexamethasone an oral loading dose of 20 mg, followed by 5 mg daily for 5–7 days. Patients weighing over 90 kg received an increased FPV dose, specifically 12 tablets (200 mg each) twice daily on Day 1, followed by 5 tablets (200 mg each) twice daily on Days 2–5. For patients with pneumonia and $SpO_2 ≤ 94\%$ or rapid worsening symptoms, dexamethasone 6 mg/day for 7–10 days is recommended, with dose adjustments for weight over 90 kg. If $SpO_2 ≤ 93\%$ or oxygen support ≥ 3 L/min is required, dexamethasone up to 20 mg/day can be given, tapering as symptoms improve over at least 7 days. For severe cases requiring high-flow nasal cannula (HFNC), non-Invasive ventilation (NIV), or mechanical ventilation, dexamethasone 20 mg/day should be administered for at least 5 days, followed by gradual tapering. If symptoms worsen, increasing the dose should be carefully weighed against the risk of infection.

The initial cohort of patients was divided into three treatment groups: (1) FPV with Dexa; (2) FPV alone; and (3) ST, based on the treatment records. In the favipiravir groups (FPV and FPV with Dexa), favipiravir was initiated within 1 day of admission for a maximum of 14 days. Supportive therapies were also administered in the ST group and were permitted across all groups.

In order to simulate random allocation, we generated three clones for each eligible patient and assigned each of them to each treatment strategy, regardless of the original treatment received. This means that the dataset was tripled, and all baseline prognostic characteristics were assumed to be balanced. We assumed that a at the time of randomization, all patients were assumed to have an equal probability of being assigned to the FPV with Dexa, FPV, or ST group (ideally at the time of BH admission).

In each treatment group, follow-up times for cloned patients who deviated from the original treatment plan were artificially censored. For instance, in a patient who originally received FPV with Dexa, the clones allocated to FPV and ST were censored at the time of FPV with Dexa prescription; however, the clone assigned to FPV with Dexa was followed up afterward. Likewise, in patients who originally received FPV, their clones who were allocated to FPV with Dexa and ST were subsequently censored; for patients who had not yet received favipiravir with dexamethasone or favipiravir alone on day 1 (originally received ST) the clones allocated to FPV with Dexa and FPV were censored on day 1, but follow up time in the ST group continued. This process is illustrated in Fig 1.

### Follow-up and endpoints

The primary outcome was in-hospital mortality within 30 days of hospitalization, starting from the date of admission at BH. All patients were observed until hospital day 30, and mortality events or discharge, whichever came first, were recorded. To correct for immortal time bias, the event was counted only for patients who received their originally assigned treatment and remained uncensored.

### Missing data handling

Missing values in prognostic factors were addressed through imputation using the k-Nearest Neighbor (kNN) imputation based on a variation of the Gower distance [26]. After excluding classification factors (age, gender, symptomatic status, favipiravir treatment status, dexamethasone treatment status), only prognostic factors were included in the imputation procedures, while outcome variables (time and mortality status) were excluded. Imputation was conducted in R using a

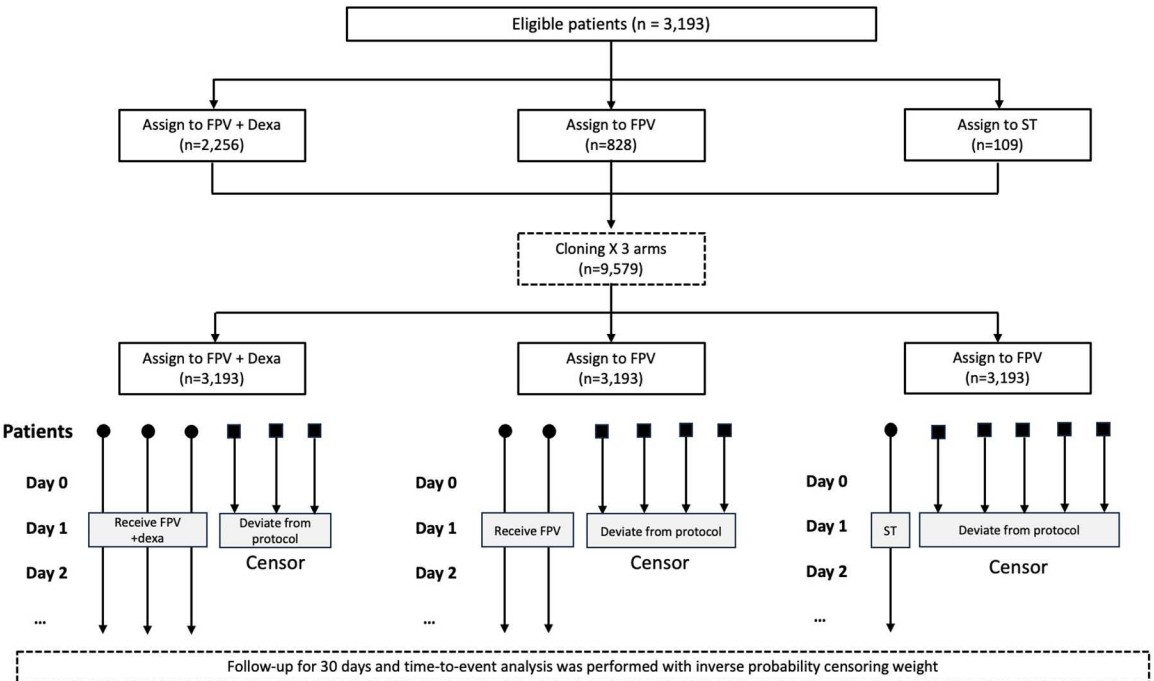

**Fig 1. Study design using target trial emulation with cloning and censoring.** A total of 3,193 eligible patients were initially assigned to one of three treatment groups: favipiravir with dexamethasone (FPV with Dexa), favipiravir (FPV) alone, or symptomatic treatment (ST). The cloning process expanded each patient across the three treatment arms (n = 9,579), ensuring comparability. Patients who deviated from their assigned protocol were censored, and follow-up was conducted for 30 days. Time-to-event analysis was performed using inverse probability of censoring weights to estimate treatment effects.

kNN function from the VIM package with default arguments setting at 5 neighbors, median function for numerical data, and maximum function for categorical data. The imputed variables were then merged with outcome variables, and the data was used for model derivation.

## Statistical analysis

All statistical analyses were performed using Stata 17 (StataCorp, College Station, TX, USA). Categorical data were described using frequencies and percentages. For numerical data, means and standard deviations or medians and inter-quartile ranges were used, as appropriate.

As the decision to prescribe favipiravir or dexamethasone might be influenced by various demographic or clinical variables often associated with the outcome, artificial censoring during follow-up could introduce selection bias. To address this issue, we employed inverse probability of censoring weighting (IPCW). This method weighted patients remaining in the risk set to maintain the comparability of the treatment strategies over the grace period and follow-up. We then predicted the probability of censoring for each treatment strategy using a treatment-specific Cox proportional hazards model.

Based on clinical knowledge, the Cox model included all potential prognostic factors and confounders. The weight model included age over 60, gender, time from PCR to admission, respiratory support, cerebrovascular disease, cardiovascular diseases, cirrhosis, chronic kidney diseases, chronic obstructive pulmonary diseases, diabetes, hypertension, immunodeficiency, obesity, body temperature, heart rate, BMI, systolic and diastolic blood pressure, oxygen saturation, and vaccination status. The censoring weights were calculated as the inverse of the predicted probabilities, estimated separately for each treatment group. Weights were truncated at the 99th percentile to remove extreme values, thereby mitigating the risk of violating the positivity assumption.

We used the standardized difference (STD) to assess the degree of differences in patient characteristics across the three groups (three comparison pairs). A significant difference between groups was defined as an absolute STD greater than 10% [27]. Any variables that remained imbalanced after weighting were adjusted for in the weighted analysis model to ensure double robustness.

The restricted mean survival time (RMST) difference was employed to evaluate the mean survival times between the three groups for the primary study endpoint, in-hospital death. RMST provides a robust alternative for comparing survival outcomes when the proportional hazards assumption is violated. Unlike Cox models, RMST does not rely on the proportionality of hazards, making it more suitable for real-world clinical data with varying treatment effects over time [28]. We performed a weighted flexible parametric survival regression to model the log cumulative hazard. This model used one degree of freedom for the baseline hazard distribution and one degree of freedom for the time-treatment interaction to estimate the RMST for each group.

## Results

### Participants

A total of 18,184 patients diagnosed with COVID-19 infection admitted to BH in April 2021, of whom 3,193 moderate and moderate-to-severe cases were included. Fig 2 describes in detail the reasons for exclusion. The average age of eligible patients was 50.90 years (SD 14.77), and male gender was slightly lower than female (44.16% vs. 55.84%). Almost half of the patients had at least one comorbidity. Approximately 76% of patients received at least one dose of any COVID-19 vaccine, whereas only 20% and 3% received a second and third dose, respectively. A total of 44.54% of patients required oxygen support including oxygen cannula, oxygen mask, and oxygen high flow.

A total of 3,193 eligible cases were treated with one of the three strategies—FPV with Dexa (2,256 patients, 70.65%), FPV alone (828 patients, 25.93%), and ST (109 patients, 3.41%). Table 2 illustrates demographics, underlying conditions, vital signs, immunization status, and respiratory support for each treatment strategy. Patients in the FPV with Dexa group

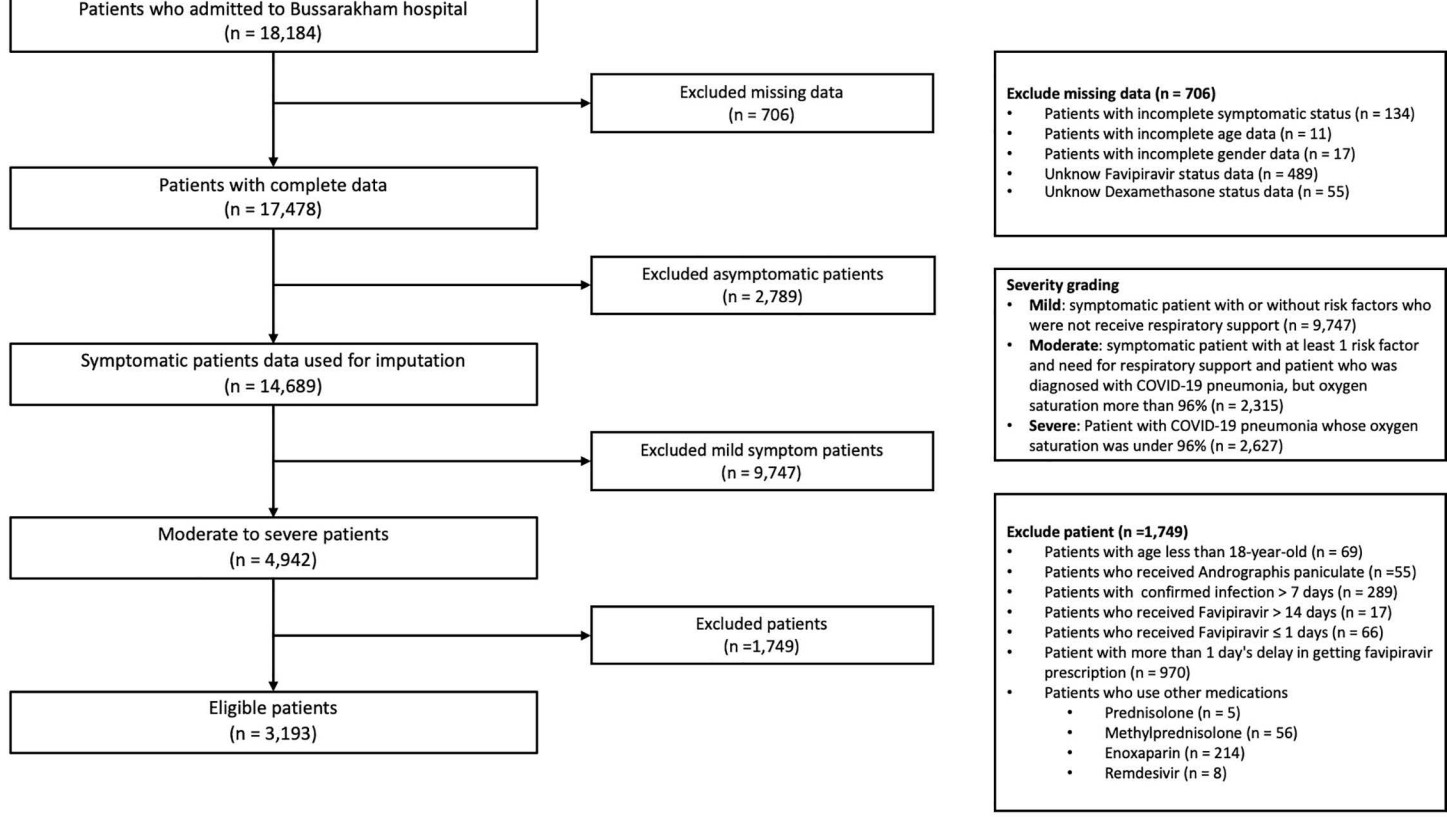

**Fig 2. Exclusions of participants.** From all patients with COVID-19 admitted to Bussarakham Field Hospital (n = 18,184), asymptomatic patients (n = 2,789) and those with mild severity (n = 9,747) were excluded. The remaining eligible patients were 3,193.

were older, had more comorbidities (i.e., diabetes mellitus, cardiovascular disease, hypertension, and obesity). A higher number of patients who obtained oxygen cannula as initial respiratory support was observed in the FPV with Dexa group, where patients were also older than those in the other two groups.

## Standardized difference pre and post cloning-censor weighting

After weighting, in comparison between ST and FPV with Dexa, all baseline characteristics were similar (STD < 0.1) except age > 60 years, BMI, CVA, diabetes mellitus, hypertension, immunodeficiency, oxygen saturation, and initial respiratory support, especially for oxygen cannula and room air. In contrast, the comparisons between ST vs. FPV and FPV with Dexa vs. FPV showed greater imbalance, with 10 and 9 out of 25 characteristics differing, respectively. The most significantly different characteristics was the use of room air as an initial respiratory support between FPV with Dexa vs. ST (STD −0.889). Fig 3A displays the unweighted standardized difference for each character. The reference line indicate STD values at ±0.1, indicating the level of acceptable similarity between characters.

We applied cloning-censoring weighting to eliminate the imbalances in the baseline prognostic characteristics. At the end of the grace period, some baseline imbalance remained between the groups. The derived censoring weights were able to balance the differences between the three groups except for the following baseline factors, age ≥ 60, male gender, diabetes mellitus, hypertension, oxygen saturation, use of room air, oxygen cannula, and oxygen high flow (Fig 3B). Details on the standardized differences for each group are described in S1 Table.

**Table 2.** Demographic, clinical characteristics, treatment, and outcome of the original patient cohort.

| Patient characteristic | Eligible patients with moderate-to-severe symptoms (n = 3,193) | | | | | |
| --- | --- | --- | --- | --- | --- | --- |
| | Favipiravir combined with dexamethasone (n = 2,256) | | Favipiravir alone (n = 828) | | Symptomatic treatment (n = 109) | |
| | N | % | N | % | N | % |
| **Demographic** | | | | | | |
| Age, year (mean ± SD) | 52.23 | 14.34 | 48.17 | 15.26 | 44.22 | 15.23 |
| Age above 60 years | 727 | 32.23 | 208 | 25.12 | 21 | 19.27 |
| Male | 1,014 | 44.95 | 345 | 41.67 | 51 | 46.79 |
| BMI, kg/m² (mean ± SD) | 27.12 | 6.20 | 26.63 | 6.07 | 25.87 | 7.26 |
| Duration from PCR to admission, day (median, IQR) | 2 | 1,4 | 3 | 1,4 | 3 | 1,5 |
| Moderate-to-severe COVID-19 | 1,241 | 55.01 | 393 | 47.46 | 65 | 59.63 |
| **Underlying disease** | | | | | | |
| Cerebrovascular disease | 17 | 0.75 | 5 | 0.60 | 1 | 0.92 |
| Cardiovascular disease | 61 | 2.70 | 20 | 2.42 | 1 | 0.92 |
| Cirrhosis | 8 | 0.35 | 0 | 0 | 0 | 0 |
| Chronic kidney disease | 19 | 0.84 | 7 | 0.85 | 1 | 0.92 |
| COPD | 13 | 0.58 | 6 | 0.72 | 1 | 0.92 |
| Diabetes mellitus | 460 | 20.39 | 91 | 10.99 | 15 | 13.76 |
| Hypertension | 745 | 32.02 | 200 | 24.15 | 26 | 23.85 |
| Immunodeficiency | 12 | 0.53 | 3 | 0.36 | 0 | 0 |
| Obesity (Body weight > 90 kg or BMI > 35 kg/m²) | 367 | 16.45 | 93 | 11.27 | 14 | 12.73 |
| **Vital sign** | | | | | | |
| Body temperature, degree Celsius (mean ± SD) | 36.52 | 0.61 | 36.45 | 0.52 | 36.50 | 0.48 |
| Heart rate, BPM (mean ± SD) | 95.07 | 15.28 | 92.86 | 14.58 | 95.40 | 15.70 |
| Systolic blood pressure, mmHg (mean ± SD) | 126.09 | 19.47 | 124.38 | 18.97 | 122.70 | 16.99 |
| Diastolic blood pressure, mmHg (mean ± SD) | 76.10 | 12.93 | 76.15 | 12.31 | 74.31 | 12.14 |
| Oxygen saturation, % (mean ± SD) | 95.50 | 3.67 | 95.95 | 2.90 | 94.00 | 5.26 |
| Oxygen saturation, % (median, IQR) | 96 | 95,97 | 97 | 96,97 | 96 | 93,97 |
| **Vaccination status** | | | | | | |
| Not receive any vaccine | 1,728 | 76.60 | 619 | 74.76 | 85 | 77.988 |
| Complete 1st dose of vaccine | 454 | 20.12 | 173 | 20.89 | 18 | 16.51 |
| Complete 2nd dose of vaccine | 74 | 3.28 | 36 | 4.35 | 6 | 5.50 |
| **First respiratory support** | | | | | | |
| Room air | 1,002 | 44.41 | 676 | 81.64 | 91 | 83.49 |
| Oxygen cannula | 1,041 | 46.14 | 127 | 15.34 | 11 | 10.09 |
| Oxygen mask | 61 | 2.70 | 9 | 1.09 | 1 | 0.92 |
| Oxygen high flow | 152 | 6.74 | 16 | 1.93 | 6 | 5.50 |

Abbreviations: SD, Standard deviation; IQR, Interquartile range; BMI, Body mass index; PCR, Polymerase chain reaction test for COVID-19; COPD, Chronic obstructive pulmonary disease.

## Survival outcome

For survival probability during the 30-day hospitalization, the weighted unadjusted Kaplan-Meier (KM) curve and the Cox proportional hazards regression adjusted for residual baseline imbalance, are displayed in Fig 4 and S2 Fig, respectively.

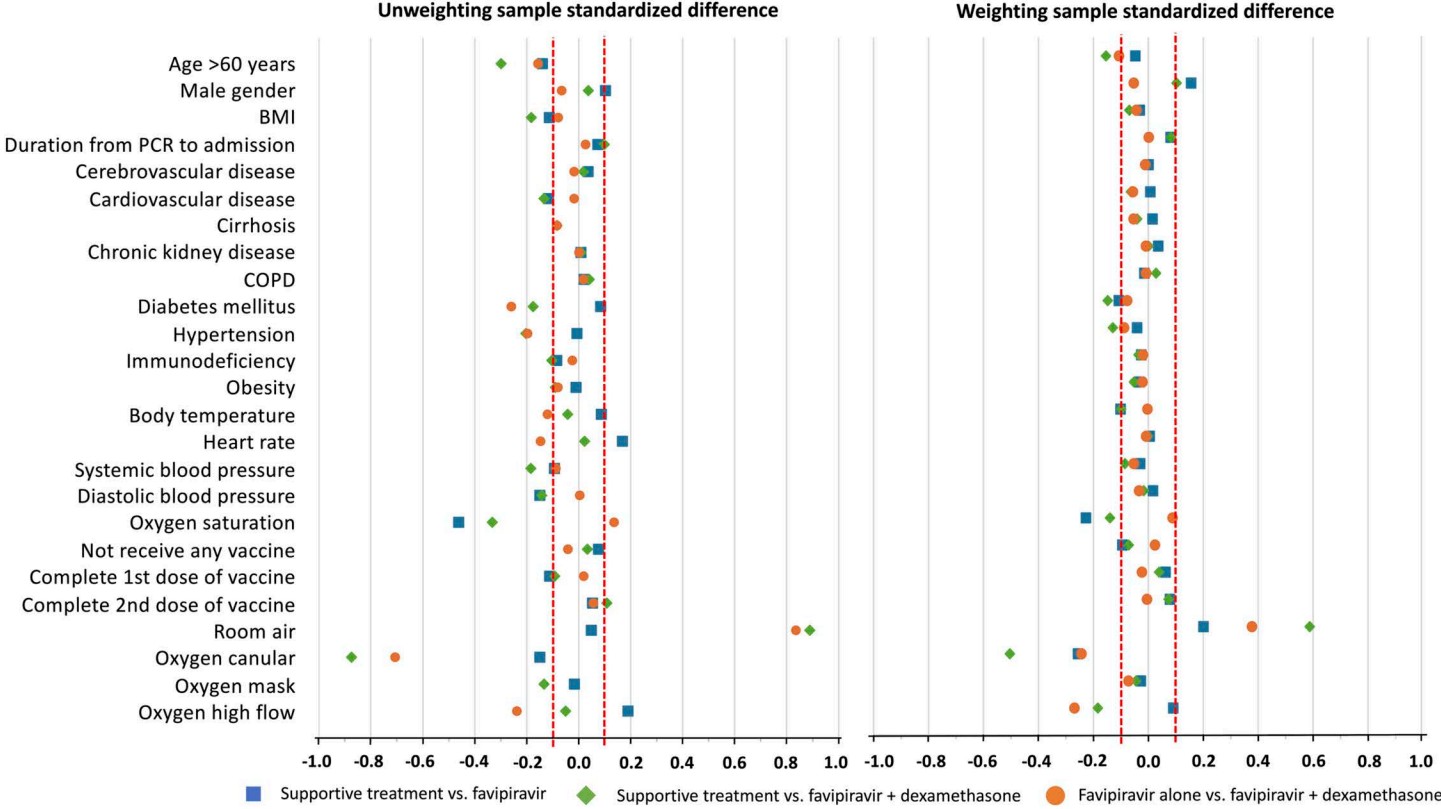

**Fig 3. Standardize differences of each baseline characteristic between treatment strategies (A) unweighted standardized difference; (B) weighted standardized difference.** The reference lines at ±0.1 indicate the threshold for acceptable similarity between characteristics. Between ST and FPV with Dexa, 13 baseline characteristics differed (STD > 0.1). Comparisons between ST vs. FPV and FPV with Dexa vs. FPV showed fewer differences, with 10 and 9 out of 25 characteristics differing, respectively.

The RMST within 30 days of hospitalization for the FPV with Dexa, FPV and ST groups were 29.68 days (95% CI: 29.52,29.84), 29.46 days (95% CI: 29.22,29.71), and 28.14 days (95% CI: 26.51,29.76), respectively. To compare the RMST difference between groups, significant differences were observed between FPV and ST groups with an adjusted RMST difference of 1.32 days (95% CI: 0.05,2.60; p-value = 0.042). In addition, a marginally significant difference was observed between FPV with Dexa and ST groups (p-value = 0.060), but not between the FPV with Dexa and FPV groups (Table 3, Fig 4).

Furthermore, to explore the robustness of the results, we performed a sensitivity analysis among moderate-to-severe patients, and those with poor prognostic factors including hypoxia (oxygen saturation <96%), male gender, age over 60 years, and lack of any COVID-19 vaccination (S2 Table). As a result, the effect of favipiravir in reducing mortality was more pronounced among patients with moderate-to-severe disease (pneumonia and oxygen saturation <96%), patients with hypoxia, and male patients either with or without dexamethasone.

## Discussion

Our study aimed to assess the efficacy of favipiravir in moderate to severe COVID-19 patients by using observational data from Thailand's largest field hospital to emulate a target trial. By emulation of a target trial where patients were hypothetically randomized to Favipiravir with dexamethasone (FPV with Dexa), Favipiravir alone (FPV alone), and symptomatic treatment (ST), our study strengthens the causal interpretation of the observed favipiravir efficacy. The findings suggest

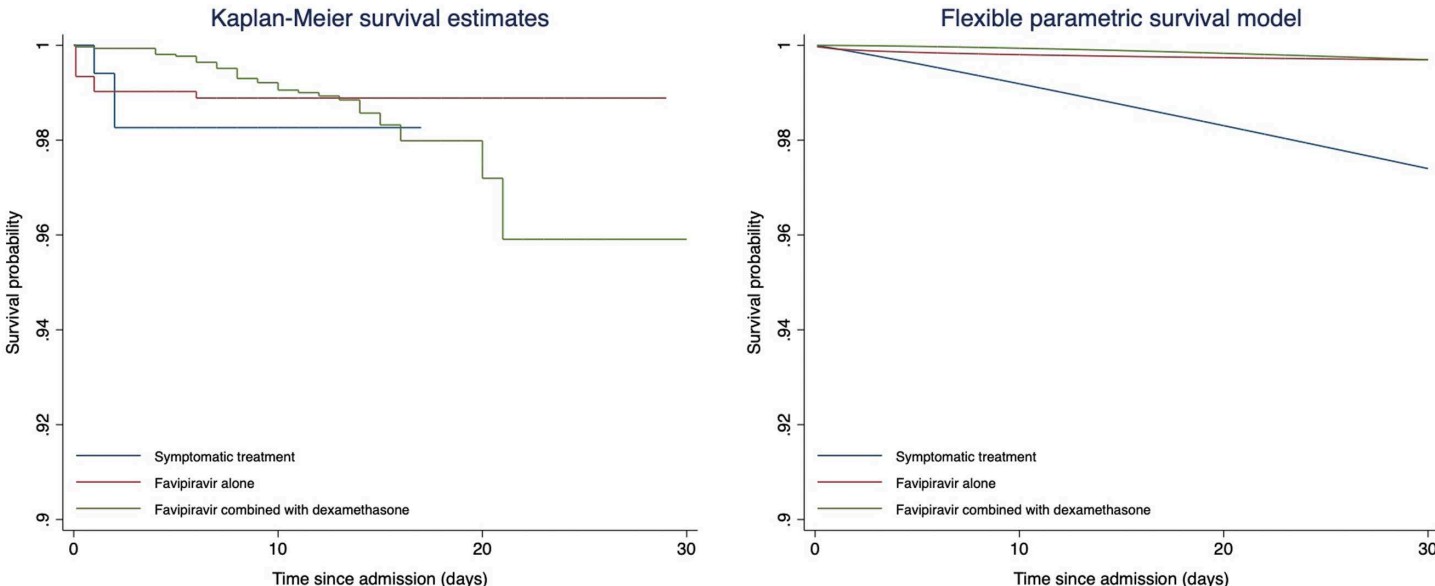

**Fig 4. Kaplan-Meier estimation of 30-day in-hospital mortality (right) and the predicted survival curve based on flexible parametric survival regression.** The Kaplan-Meier curve shows empirical survival probabilities over 30 days for patients receiving symptomatic treatment (blue), favipiravir alone (red), and favipiravir combined with dexamethasone (green). The flexible parametric model provides a smoothed survival estimate, accounting for time-dependent effects. Both models indicate that survival probabilities differ across treatment groups, with the symptomatic treatment group showing a slightly lower survival probability over time.

**Table 3. Restricted mean survival time of in-hospital mortality for each treatment strategy.**

| Treatment strategies | RMST | | Adjusted RMST Difference compared to ST | | | Adjusted RMST Difference compared to FPV alone | | |
|---|---|---|---|---|---|---|---|---|
| | Mean (days) | 95% CI | Mean (days) | 95% CI | p-value | Mean (days) | 95% CI | p-value |
| ST | 28.14 | 26.51,29.76 | – | – | – | – | – | – |
| FPV alone | 29.46 | 29.22,29.71 | 1.32 | 0.05,2.60 | **0.042** | – | – | – |
| FPV with Dexa | 29.68 | 29.52,29.84 | 1.54 | −0.06,3.15 | 0.060 | 0.22 | −0.23, 0.67 | 0.335 |

Abbreviations: CI, confidence interval; Dexa, dexamethasone; FPV, favipiravir; RMST, restricted mean survival time; ST, symptomatic treatment.

a modest survival benefit from Favipiravir-based therapies which demonstrated an average gain of 1–2 days of survival time. The results reflect clinically meaningful reductions in mortality risk over a large cohort, especially in a high-risk hospitalized population. Importantly, the direction of effects was consistent across sensitivity analyses among patients with specific prognostic factors, supporting the plausibility of a therapeutic benefit from favipiravir.

To compare survival outcomes among treatments, the emulated target trial methodology is particularly relevant in our context, as traditional observational analyses are prone to bias. First, imbalanced prognostic characteristics between treatment arms can result in baseline confounding. Second, differences in treatment initiation times can lead to immortal time bias [20,21]. Patients who were assigned to receive FPV and ended up in the treated arm must survive until the treatment initiation. This artificially contributes to an inflated beneficial effect of FPV if both symptomatic treatment and FPV groups were equally defined at time zero and followed up while FPV groups must survive until prescription and the actual follow-up begins. By emulating a trial protocol, we specifically defined a time zero of follow-up at admission for all treatment groups and allowed a grace period of one day after admission to receive FPV. Then, we cloned the patients,

tripled the dataset, and randomly assigned them, thereby balancing the baseline across all study arms. The cloned data were censored, and the survival time analysis was weighted by censoring probability. Although propensity score matching studies have also been conducted to evaluate the efficacy of favipiravir [29–31], our study provides additional bias adjustment through cloning and censoring. From our analysis, the baseline BMI, duration from PCR to admission, underlying conditions except DM and hypertension, and vaccination status were well balanced after weighting, while the remaining were later adjusted for in the RMST model. Although emulating a target trial may not fully replace an actual randomized controlled trial, but in a pandemic setting, this approach offers a rigorous way to extract causal insights from real-world data with minimized baseline imbalance and immortal time bias when RCTs are limited or delayed [32].

A number of clinical trials and observational studies have reported controversial results regarding mortality rates among moderate to severe COVID-19 patients treated with favipiravir (S3 Table). The largest randomized trial to date, the open-label PIONEER trial with 503 patients, reported no significant difference in 28-day mortality between Favipiravir and standard care overall (10% vs 14% mortality; HR 0.74, p = 0.24) [33]. However, that trial hinted at potential subgroup effects: in patients under 60 years old, favipiravir significantly improved outcomes, including a 66% reduction in the risk of progressing to mechanical ventilation or death (HR 0.34, p = 0.02). While an RCT from Shenoy et al. in Kuwait reported no significant difference in mortality [34], a study by Kamali et al. in Iran showed significantly better results of favipiravir in critically ill patients [35]. Although this study has imbalanced baseline oxygen saturation, the improvement in the FPV group, which had a mean oxygen saturation <80%, was notable. Consistently in a real-world study, though the analysis was unmatched, favipiravir monotherapy can reduce 28-day mortality risk in severe COVID-19 compared to no-antiviral treatment (relative risk = 0.72; 95% CI 0.58–0.91; p = 0.006) [19]. Our results similarly revealed that in moderate-to-severe case, patients tend to benefit from favipiravir treatment compared to standard care or symptomatic treatment, especially in patients with hypoxia, but not older patients.

Furthermore, the inclusion of dexamethasone in our combination arm, FPV with Dexa, warrants discussion. Dexamethasone is a cornerstone of COVID-19 therapy following the RECOVERY trial, which demonstrated a significant mortality reduction. Patients on mechanical ventilation or oxygen therapy had substantially lower death rates with dexamethasone (e.g., 29.3% vs 41.4% among ventilated patients) whereas patients not needing oxygen had no benefit and a non-significant trend with steroids [15]. Our results demonstrate an added survival benefit when dexamethasone is combined with favipiravir providing a trend in prolonging survival time which was more pronounced in patients with pneumonia or hypoxia. Similar results have been observed in several trials [36,37]. However, the effect of dexamethasone in our study was not large enough to show a significant improvement compared to FPV alone. This might be because favipiravir's effect might hinder the anti-inflammatory action of dexamethasone.

The observed effects of favipiravir and dexamethasone can be understood in light of their pharmacological actions. Favipiravir has broad activity against RNA viruses by inhibiting the viral RNA-dependent RNA polymerase of SARS-CoV-2 and was one of the first antivirals repurposed for COVID-19. By reducing viral load, favipiravir may shorten the period of high viral replication and potentially mitigate the direct cytopathic effects of the virus [38]. This results in faster resolution of fever and pneumonia, and a lower likelihood that the infection triggers a severe inflammatory cascade [6,39,40]. Importantly, favipiravir's efficacy appears to be highly dependent on timing [16,41]. In our moderate cases, favipiravir alone likely helped by curbing the virus during the acute phase, as supported by the modest survival improvement and aligns with reports of quicker clinical improvement in other studies [39,42]. In more severe cases, any antiviral effect of favipiravir would be adjunctive, and its combination with other supportive therapies has shown more favorable outcomes for COVID-19 patients [39]. This is biologically plausible because the ongoing viral replication may persist in some patients driving inflammation and a marked increase in inflammatory markers [14,43,44]. This is where Dexamethasone plays a crucial role. Dexamethasone is a potent glucocorticoid that downregulates the production of pro-inflammatory cytokines, collectively reducing the inflammation-mediated lung injury that characterizes acute respiratory distress syndrome in severe COVID-19 [44–46]. It can therefore be inferred that dexamethasone benefits the critically ill or hypoxic patients [15,36],

but should not be used in mild cases who were not receiving respiratory support where possible harm can be observed due to impaired viral clearance from immunosuppression [15,36].

With that being said, current variants and outpatient therapies have shifted the treatment landscape. Our study was conducted in the unique context of Thailand's largest field hospital during the country's third wave of the COVID-19 pandemic (May–September 2020), when the access to remdesivir was limited and favipiravir was used as the frontline antiviral. At that time, the Thai government primarily imported favipiravir, and secured its own supply by producing generic favipiravir tablets domestically. Therefore, favipiravir became the most widely used antiviral for symptomatic COVID-19 patients with risk factors [47]. Although available, remdesivir was reserved for select cases based on physician discretion, particularly for pregnant women in their first trimester to avoid teratogenic effects and for patients unable to take oral medications due to absorption issues. This context reflects a setting where treatment decisions were influenced by medication availability and evolving clinical practices. While these circumstances may differ from later phases of the pandemic, particularly with the emergence of new variants, expanded vaccination coverage, and broader access to alternative antivirals, they represent real-world conditions faced by resource-limited settings during critical surges. These setting-specific factors may limit the generalizability of our findings to other populations or healthcare systems. However, the use of an emulated target trial framework in our analysis enhances internal validity and supports causal inference. Additionally, the observed survival benefit of favipiravir, particularly in hypoxic patients, is consistent with findings from other trials and observational studies. Though our results suggest a modest gain in survival time with favipiravir, the benefit is within the same range as that reported for remdesivir and other antiviral agents such as remdesivir and other antivirals such as lopinavir/ ritonavir (S2 table) [30,40,48–50]. When used in conjunction with anti-inflammatory agents like dexamethasone, it may improve outcomes. Our findings suggest that favipiravir can be a valuable alternative in settings where other antivirals are inaccessible. While it may not be the first choice for outpatients, it could be useful for severely ill hospitalized patients with low oxygen saturation.

## Limitations

Our study utilized real-world data from the largest field hospital in Thailand, with an aim to further substantiate the efficacy of favipiravir in the treatment of COVID-19. However, our study carries some limitations. First, the exclusion of incomplete data and the use of strict inclusion criteria may have introduced selection bias, although this approach mimics the clinical trial environment. Second, the study was conducted during the early phase of the pandemic, when few patients were vaccinated, and management practices differed. Therefore, the generalizability of our findings warrants careful considerations in the context of the current endemic situation. Third, due to the retrospective nature of our study and the field hospital setting, the completeness of certain baseline characteristics was inherently limited. This includes missing data on the WHO COVID-19 severity scale, CRP levels, anti-interleukin treatment, disease progression, progression to mechanical ventilation, and time to discharge. Although we had data on respiratory support, we lacked details on vasopressor use and ECMO. However, as all patients in this study were hospitalized, the WHO ordinal scale would have ranged from 4 to 10. Fourth, management strategies during the pandemic also introduced variability in the exact time from symptom onset to hospital admission. We therefore replaced the time from symptom onset to admission with the PCR test date to admission, as this variable was more reliable, widely available across all admitted patients, and free from recall bias. Our study focused mainly on mortality rather than disease progression or ICU admission rates, which are commonly used to describe the disease trajectory in moderate COVID-19 cases. Instead, we emphasized the critical endpoint of in-hospital mortality, which remains a major concern even after ICU admission. Future studies with more comprehensive clinical outcomes will be necessary to better assess treatment impact across different severity levels. The small sample size in the symptomatic treatment group may have reduced statistical power, introducing instability and variability in treatment effect estimates, and potentially affecting the assumptions of exchangeability and positivity. To address this, we employed inverse probability of censoring weighting (IPCW), adjusted for imbalance factors, and conducted sensitivity

analyses. However, limited sample size may still challenge the validity of these assumptions. Additionally, despite adjusting for a comprehensive set of covariates, residual unmeasured confounding cannot be excluded, which may compromise exchangeability. Future studies with larger samples, greater statistical power, and more detailed covariate data are needed to clarify early survival outcomes and strengthen causal inference. Lastly, our results, derived from an observational real-world setting, should be validated by prospective controlled trials to confirm their applicability.

## Conclusions

Our emulated target trial suggests that favipiravir, especially when combined with dexamethasone, may offer modest survival benefits for hospitalized patients with moderate to severe COVID-19, particularly among patients with hypoxia. Our analysis contributes to the growing evidence in enhancing causal relationship of favipiravir in a real-world setting. While the benefits may not be substantial, favipiravir remains a practical oral antiviral in resource-limited environments. These findings support its use as an adjunct to corticosteroids when intervention is feasible, although further controlled studies are needed to strengthen clinical recommendations.

## Supporting information

**S1 Fig. Treatment order of Bussarakham Field hospital.**
(TIF)

**S2 Fig. Cox proportional hazard regression corrected for residual baseline imbalance.**
(TIF)

**S1 Table. Standardized difference (before vs after weighting).**
(DOCX)

**S2 Table. Sensitivity analysis among patients with more severe characteristics.**
(DOCX)

**S3 Table. Studies investigated mortality outcomes of favipiravir in hospitalized moderate to critical COVID-19 patients.**
(DOCX)

## Acknowledgments

This research was partially supported by Chiang Mai University and Faculty of Medicine, Chiang Mai University.

## Author contributions

**Conceptualization:** Lalita Lumkul, Krittai Tanasombatkul, Thotsaporn Morasert, Phichayut Phinyo.

**Data curation:** Lalita Lumkul, Krittai Tanasombatkul, Thotsaporn Morasert, Phichayut Phinyo.

**Formal analysis:** Lalita Lumkul, Krittai Tanasombatkul, Thotsaporn Morasert, Phichayut Phinyo.

**Investigation:** Lalita Lumkul, Krittai Tanasombatkul, Phongsak Nitikaroon, Thotsaporn Morasert, Phichayut Phinyo.

**Methodology:** Lalita Lumkul, Krittai Tanasombatkul, Phongsak Nitikaroon, Thotsaporn Morasert, Phichayut Phinyo.

**Project administration:** Thotsaporn Morasert, Phichayut Phinyo.

**Resources:** Krittai Tanasombatkul, Phongsak Nitikaroon, Thotsaporn Morasert.

**Software:** Krittai Tanasombatkul, Phongsak Nitikaroon.

**Supervision:** Thotsaporn Morasert, Phichayut Phinyo.

**Visualization:** Phichayut Phinyo.

**Writing – original draft:** Lalita Lumkul, Krittai Tanasombatkul.

**Writing – review & editing:** Lalita Lumkul, Krittai Tanasombatkul, Phongsak Nitikaroon, Thotsaporn Morasert, Phichayut Phinyo.

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
