## [Decision Letter · Decision Letter 0]

3 Dec 2024

PONE-D-24-51078In-hospital mortality outcome of favipiravir in patients with moderate to severe COVID-19 infection: an emulated target trial using real-world data from the largest field hospital in ThailandPLOS ONE

Dear Dr. Phinyo

Thank you for submitting your manuscript to PLOS ONE. After careful consideration, we feel that it has merit but does not fully meet PLOS ONE’s publication criteria as it currently stands. The reviewers raised several points during revision and we invite you to address these points and provide a revised version of your manuscript.

The study from Lalita Lumkul et al. describes an emulated trial for the evaluation of Favipiravir efficacy in the treatment of symptomatic COVID-19 patients in Busarakam field hospital, in Thailand. The authors concluded that Favipiravir, alone or in combination with dexamethasone, did not significantly improve the outcome when compared to standard treatment.

This work may be of interest for publication in PLOS ONE but needs improvement. I invite you to pay attention to the reviewers' comments, which I believe to be very valuable to improve your interesting work. Suggestions from reviewers regard the contextualisation of your study as well as the methodology description, which I feel has been more criticised and I invite you to revise in order to address the points that were raised.

We look forward to receiving your revised manuscript.

Kind regards,

Dr. Chiara Bertagnin

Academic Editor

PLOS ONE

Journal Requirements:

Reviewers' comments:

Reviewer's Responses to Questions

**Comments to the Author**

1. Is the manuscript technically sound, and do the data support the conclusions?

Reviewer #1: Yes

Reviewer #2: No

2. Has the statistical analysis been performed appropriately and rigorously? 

Reviewer #1: Yes

Reviewer #2: Yes

3. Have the authors made all data underlying the findings in their manuscript fully available?

Reviewer #1: Yes

Reviewer #2: No

4. Is the manuscript presented in an intelligible fashion and written in standard English?

Reviewer #1: Yes

Reviewer #2: Yes

5. Review Comments to the Author

Reviewer #1: Dear Authors

Thank you for your effort in this study.

The manuscript titled "In-hospital mortality outcome of favipiravir in patients with moderate to severe COVID-19 infection: an emulated target trial using real-world data from the largest field hospital in Thailand" evaluates the efficacy of favipiravir, with or without dexamethasone, in reducing in-hospital mortality among moderate to severe COVID-19 patients. Using a target trial emulation framework on data from over 3,000 patients in Thailand's largest field hospital, the study finds no significant improvement in 30-day survival outcomes for patients treated with favipiravir (alone or combined with dexamethasone) compared to those receiving standard symptomatic care.

The study highlights limitations such as residual confounding and the retrospective design, yet it contributes to the growing evidence questioning favipiravir’s role in treating severe COVID-19. It underscores the need for more rigorous prospective trials and suggests that favipiravir may not be essential in clinical practice for such cases.

While the study is well-conceived and executed, several areas require clarification, expanded discussion, and improved presentation to maximize its impact and clarity.

My comments:

1- Abstract

Strengths:

- The abstract is concise and provides a clear overview of the study’s aims, methods, results, and conclusions.

- It effectively communicates the novelty of the approach (emulated target trial) and its primary finding (no significant difference in mortality outcomes across treatment groups).

Criticisms:

- The p-values mentioned for the survival outcomes (e.g., p-value 0.479, 0.366) do not convey a strong interpretative value to general readers and could be omitted for brevity.

- The conclusion could better emphasize the implications of these findings for clinical guidelines, particularly in contexts where favipiravir is still widely used.

Suggestions for Improvement:

- Replace some numerical details with a more narrative style to improve accessibility (e.g., “no significant mortality difference was observed” instead of detailed p-values).

- Add a statement on the study’s real-world relevance to emphasize its significance.

2- Introduction

Strengths:

- Provides a detailed background on the role of favipiravir and its widespread use during the pandemic.

- Highlights the gap in evidence for its efficacy in moderate to severe COVID-19 cases, setting a strong rationale for the study.

Criticisms:

- Redundancy in describing the historical context of favipiravir’s use and the delta wave pandemic. While informative, some details could be streamlined.

- Limited critical analysis of conflicting evidence regarding favipiravir’s efficacy from past studies.

Suggestions for Improvement:

- Condense the description of favipiravir’s history to focus on its relevance to Thailand’s healthcare system.

- Introduce a clearer comparison of existing studies, highlighting where this study fills a knowledge gap.

3- Methods

Strengths:

- The study design is robust, with careful consideration of biases such as immortal time bias.

- Comprehensive description of statistical techniques, including inverse probability of censoring weighting (IPCW) and restricted mean survival time (RMST) analysis.

Criticisms:

- The description of cloning and censoring might confuse readers unfamiliar with advanced epidemiological methods.

- Simplification or clarification is necessary.

- Potential residual confounding due to unmeasured variables (e.g., adherence to treatment protocols) is not addressed.

Suggestions for Improvement:

- Add a brief supplementary explanation or diagram illustrating the cloning-censoring process.

- Acknowledge the potential for residual confounding and discuss its impact on results.

4- Results

Strengths:

- Results are well-organized with clear presentation of key findings through Kaplan-Meier curves and RMST analyses.

- Comprehensive demographic data provide context for interpreting the findings.

Criticisms:

- Baseline imbalances between treatment groups (e.g., age and vaccination status) may bias the findings, and while addressed statistically, their potential impact is not discussed adequately.

- The exclusion criteria (e.g., patients with delayed treatment initiation) could introduce selection bias, limiting generalizability.

Suggestions for Improvement:

- Discuss the extent to which baseline imbalances, even after weighting, could affect the study’s conclusions.

- Include sensitivity analyses or stratified results to explore the robustness of findings across subgroups (e.g., age or comorbidity status).

5- Discussion

Strengths:

- The discussion places findings within the context of existing literature, effectively contrasting with other observational studies and randomized controlled trials.

- The limitations section is thorough, acknowledging the retrospective nature of the data and the early-pandemic context.

Criticisms:

- The discussion on favipiravir’s ineffectiveness could be expanded to consider potential mechanistic reasons for these findings, such as pharmacokinetic or pharmacodynamic limitations.

- While limitations are acknowledged, no actionable recommendations for future research are provided beyond the need for prospective trials.

Suggestions for Improvement:

- Add a brief discussion of potential pharmacological explanations for the lack of efficacy.

- Suggest specific future research directions, such as trials investigating favipiravir in specific subpopulations or combination therapies.

6- Figures and Tables

Strengths:

- Figures (e.g., Kaplan-Meier curves) and tables are clear and provide sufficient detail for readers to assess the results.

Criticisms:

- The font size in some figures is small, making them difficult to read.

- Captions are descriptive but could provide more context on the statistical methods applied.

Suggestions for Improvement:

- Improve figure readability by increasing font size and ensuring high resolution.

- Revise captions to briefly mention the key takeaways from the figures.

7- Ethical Considerations

Strengths:

- The ethical approval process and waiver for informed consent are clearly described.

- Data availability is well-documented, adhering to open science principles.

Criticisms:

- The manuscript does not address potential ethical concerns regarding the use of data collected in emergency settings, such as the implications of waiving informed consent.

Suggestions for Improvement:

- Briefly discuss the ethical trade-offs of waiving consent in retrospective studies and the safeguards implemented to protect patient data.

8- Conclusion

Strengths:

- The conclusion effectively summarizes the findings and aligns them with the study’s objectives.

Criticisms:

- The conclusion does not sufficiently emphasize the study's implications for global COVID-19 treatment guidelines.

- Overemphasis on favipiravir’s lack of efficacy without suggesting contexts where it might still be useful.

Suggestions for Improvement:

- Reframe the conclusion to include a broader discussion of the findings’ relevance to current treatment paradigms and resource-limited settings.

Thank you

Reviewer #2: I would like to express my gratitude to the editors for providing me with the opportunity to review this intriguing manuscript.

Comment 1:

The manuscript does not describe whether all the assumptions for target trial emulation (consistency, positivity, and exchangeability) were explicitly tested or validated.

The creation of clones for patients assumes that treatment assignment does not influence subsequent outcomes—a critical assumption that may not be fully met in real-world settings.

Comment 2

The description of how censoring weights were calculated is not detailed. Lack of transparency about how weights were derived can affect reproducibility and credibility of results

Comment 3:

The manuscript identifies moderate imbalances in baseline characteristics across treatment groups, even after weighting (Table S1), including variables such as age, comorbidities, and initial respiratory support. These imbalances may undermine the validity of causal inferences and could necessitate sensitivity analyses to adjust for these covariates and reduce bias.

Additionally, key factors such as the WHO ordinal scale, hypoxemia on admission, and the duration from symptom onset to hospital admission should also be accounted for in the analysis. Importantly, the duration from PCR testing to admission is not a valid measure and does not adequately represent the likelihood of increased disease severity.

Comment 4:

The absence of key baseline characteristics, such as obesity or morbid obesity (defined as BMI >30 kg/m², as opposed to using body weight >90 kg, which is insufficient—please review the proper definition of obesity), the time from symptom onset to hospital admission, the WHO COVID-19 severity scale, baseline hypoxemia, and the precise duration from symptom onset to hospitalization (e.g., distinguishing between 3 days and 7 days, as these may have significantly different implications) raises concerns about potential bias and the validity of the study. Furthermore, the lack of data on CRP levels, the percentage of pulmonary involvement on chest X-ray, and progression to the need for mechanical ventilation further limits the robustness of the findings and may compromise the reliability of the conclusions drawn.

Comment 5:

The authors did not differentiate between moderate and severe COVID-19 cases, which is a critical oversight. For moderate COVID-19, outcomes such as disease progression and hospital admission are more appropriate measures, rather than mortality. Conversely, severe COVID-19 should be evaluated using metrics such as the need for mechanical ventilation, ICU admission or transfer to a nearby tertiary hospital, ICU length of stay, development of ARDS, respiratory failure, and mortality.

This lack of distinction may stem from an inappropriate study design, which fails to account for the differing clinical trajectories and outcome measures relevant to moderate and severe cases.

Comment 6

The model used to predict the probability of censoring includes "all potential prognostic factors and confounders," but the manuscript lacks justification for the variables included and excluded. These were the key confounders that led to compromise results.

Comment 7

Restricted Mean Survival Time differences are reported, the manuscript does not clearly indicate if assumptions for parametric survival models were met or if alternative models were considered.

Comment 8

The very small sample size in the symptomatic treatment group may result from inadequate statistical power. A limited sample size reduces the reliability of estimated effects because statistical power depends on the number of actual observed events rather than an artificially balanced dataset. This limitation likely contributed to the low number of observed events in the standard treatment group.

Moreover, the cloning approach, which creates clones for all treatment groups, has inherent challenges. Clones assigned to groups that do not align with the original treatment are censored early, leading to shorter follow-up times and fewer data points available to estimate treatment effects. This issue is particularly pronounced for smaller sample sizes, as it results in unstable or extreme weights for clones in the symptomatic treatment group. Such instability inflates variance in the analysis and further reduces the statistical power, undermining the robustness of the findings.

As a result, the symptomatic treatment group may lack sufficient representation for a robust and reliable comparison. To address this, more sophisticated weighting methods should be employed to improve stability and mitigate the issues associated with extreme weights. Additionally, a thorough sensitivity analysis should be conducted to assess the robustness of the results and ensure that the findings are not unduly influenced by the limitations of the current approach.

Comment 9

The in-hospital mortality data should be further classified to enhance clarity. I recommend that the authors analyze 7, 14-day and 30-day mortality rates to provide a more detailed perspective and facilitate a clearer discussion of the survival analysis.

Comment 10:

The handle with missing data should be mentioned.

Comment 11:

The authors should address the generalizability of their findings, given that the study was conducted within a specific context—a large field hospital in Thailand—during a particular wave of the COVID-19 pandemic. This unique setting and time frame may limit the applicability of the results to other populations, healthcare settings, or pandemic phases.

Comment 12:

The use of anti-inflammatory treatments, particularly anti-IL-6 therapies or the specific dose of dexamethasone, should be clearly detailed in the baseline characteristics, methods, and subgroup analysis

Comment 13:

A subgroup survival analysis stratified by moderate and severe COVID-19 cases is essential to clarify the distinct outcomes in these groups and to enhance the generalizability of the findings

Comment 14:

Line 95-102: More literature should be reviewed about the outcome of favipiravir in each severity, which was needed in the introduction. Many global reports have been published with different severity, including in Thailand, that the authors should provide citations.

Comment 15:

Line 138-145: To facilitate easier understanding for the audience, the authors should provide a clear description of the phase of the COVID-19 outbreak during the study period in the specific setting.

Comment 16:

I have come across several studies that explore increasing the dose of favipiravir (FPV) for obese patients. However, this important aspect is not mentioned or addressed in the current study.

Comment 17

The authors should discuss why severe COVID-19 cases in the study did not receive remdesivir as the antiviral treatment, particularly given its established role in managing severe cases. To the best of my knowledge, favipiravir has not been approved for use in severe COVID-19 cases since the onset of the pandemic. Addressing this issue would clarify the rationale behind the choice of antiviral treatment and provide a more comprehensive context for interpreting the study’s findings.

Comment 18

The definition provided for a moderate COVID-19 case, “The moderate case is a symptomatic patient with at least one risk factor and the need for respiratory support,” is incorrect and requires significant revision. A proper definition, supported by appropriate citations, should be included. For example, according to the WHO clinical management guidelines, moderate COVID-19 is typically defined as a symptomatic patient with evidence of lower respiratory tract infection (e.g., clinical signs of pneumonia) but without the need for supplemental oxygen. The authors should revise this statement to align with established guidelines and provide a clear reference..

Comment 19

The authors should provide an explanation for the variation in treatment regimens among patients, specifically why some received favipiravir (FPV) alone, others received a combination of FPV and dexamethasone, and why certain patients were managed with only symptomatic treatment. Clarifying these treatment decisions, including the clinical rationale, patient characteristics, or resource availability that influenced them, is essential to understanding the study design and its implications..

Comment 20:

In Table 2, it is evident that the FPV-alone and symptomatic treatment groups included a higher proportion of patients without hypoxemia, indicating that the majority of moderate COVID-19 cases in these groups were less severe. Conversely, more than 50% of patients in the FPV/dexamethasone group required oxygen supplementation, compared to less than 18% in both the FPV-alone and symptomatic treatment groups. This disparity suggests that the FPV-alone and symptomatic treatment groups had lower disease severity, which could introduce bias when evaluating the treatment effectiveness across these groups.

Comment 21

Please check the standard abbreviations in Table 2

Comment 22

Line 365-373: These statements were invalid. While dexamethasone has demonstrated clear benefits in severe COVID-19 cases, the data in the study indicate that 44% of patients in the dexamethasone group had non-severe COVID-19. This means that only 56% of patients with severe COVID-19 might have benefitted from dexamethasone, raising questions about its appropriate use in this cohort.

This reinforces my recommendation for the authors to analyze moderate COVID-19 cases separately from severe COVID-19 cases. Evaluating severe COVID-19 in isolation may downplay the role of antiviral agents, as these cases are often beyond the viral replication phase where antiviral drugs are most effective. In contrast, moderate COVID-19 cases, which are typically in the viral replication phase, could demonstrate a more pronounced benefit from antiviral treatments.

Comment 23:

I recommend that the authors engage in a more in-depth discussion to highlight the novelty of their research for the audience. This should include a detailed explanation of the authors' specific methodology, emphasizing how it differs from or improves upon existing approaches

Comment 24:

The Kaplan-Meier (KM) estimation graph reveals a surprising drop in survival probability within the FPV-alone and symptomatic treatment groups, where the majority of cases are presumed to involve moderate COVID-19. This unexpected result raises questions about potential underlying factors influencing survival in these groups. To provide greater clarity, I strongly recommend that the authors separate moderate COVID-19 cases from severe COVID-19 cases in their analysis. This stratification would allow for a more accurate interpretation of survival outcomes and ensure that the observed trends are not confounded by differing disease severities.

Comment 25:

For covariates with very small, standardized differences after weighting (e.g., age, BMI, comorbidities), the results can be interpreted with higher confidence as these variables are well-balanced between groups.

Overall, I recommend the authors consult the medical statisticians and include them in coauthors to clarify and help improve residual imbalances and further enhance the validity of your study results.

6. PLOS authors have the option to publish the peer review history of their article (what does this mean? ). If published, this will include your full peer review and any attached files.

**Do you want your identity to be public for this peer review?** For information about this choice, including consent withdrawal, please see our Privacy Policy .

Reviewer #1: **Yes: ** Ahmed Hosny Hassan

Reviewer #2: No

---

## [Author Response · Author response to Decision Letter 1]

29 Apr 2025

Dear Editor and reviewers

We very much appreciate the excellent and thorough review of our manuscript entitled “In-hospital mortality outcome of favipiravir in patients with moderate to severe COVID-19 infection: an emulated target trial using real-world data from the largest field hospital in Thailand”. We have addressed all comments, point-by-point, as attached.

Sincerely,

Phichayut Phinyo, MD, PhD

Department of Biomedical Informatics and Clinical Epidemiology,

Faculty of Medicine, Chiang Mai University,

110 Intawaroros Road, Si Phum, Muang, Chiang Mai, Thailand 50200,

Reviewers' comments:

Reviewer's Responses to Questions

Comments to the Author

1. Is the manuscript technically sound, and do the data support the conclusions?

Reviewer #1: Yes

Reviewer #2: No

2. Has the statistical analysis been performed appropriately and rigorously?

Reviewer #1: Yes

Reviewer #2: Yes

3. Have the authors made all data underlying the findings in their manuscript fully available?

Reviewer #1: Yes

Reviewer #2: No

4. Is the manuscript presented in an intelligible fashion and written in standard English?

Reviewer #1: Yes

Reviewer #2: Yes

5. Review Comments to the Author

Reviewer #1: Dear Authors

Thank you for your effort in this study.

The manuscript titled "In-hospital mortality outcome of favipiravir in patients with moderate to severe COVID-19 infection: an emulated target trial using real-world data from the largest field hospital in Thailand" evaluates the efficacy of favipiravir, with or without dexamethasone, in reducing in-hospital mortality among moderate to severe COVID-19 patients. Using a target trial emulation framework on data from over 3,000 patients in Thailand's largest field hospital, the study finds no significant improvement in 30-day survival outcomes for patients treated with favipiravir (alone or combined with dexamethasone) compared to those receiving standard symptomatic care.

The study highlights limitations such as residual confounding and the retrospective design, yet it contributes to the growing evidence questioning favipiravir’s role in treating severe COVID-19. It underscores the need for more rigorous prospective trials and suggests that favipiravir may not be essential in clinical practice for such cases.

While the study is well-conceived and executed, several areas require clarification, expanded discussion, and improved presentation to maximize its impact and clarity.

My comments:

1- Abstract

Strengths:

- The abstract is concise and provides a clear overview of the study’s aims, methods, results, and conclusions.

- It effectively communicates the novelty of the approach (emulated target trial) and its primary finding (no significant difference in mortality outcomes across treatment groups).

Criticisms:

- The p-values mentioned for the survival outcomes (e.g., p-value 0.479, 0.366) do not convey a strong interpretative value to general readers and could be omitted for brevity.

- The conclusion could better emphasize the implications of these findings for clinical guidelines, particularly in contexts where favipiravir is still widely used.

Suggestions for Improvement:

- Replace some numerical details with a more narrative style to improve accessibility (e.g., “no significant mortality difference was observed” instead of detailed p-values).

- Add a statement on the study’s real-world relevance to emphasize its significance.

Response

Thank you for your suggestion.

I have revised the abstracted and information related to the reviewers’ comment as follows:

Abstract

Background: Favipiravir, an antiviral agent, has been widely used to treat COVID-19 due to potential mechanism of action despite limited evidence of its efficacy in moderate to severe cases.

Aim: This study aimed to evaluate the efficacy of favipiravir in improving in-hospital mortality outcomes in moderate to severe COVID-19 patients through an emulation of a target trial.

Methods: We emulated a target trial using observational data from Bussarakham field hospital, Thailand during May 14 to September 20, 2021. Patients were categorized into three groups: those receiving favipiravir with dexamethasone (FPV with dexa), favipiravir alone (FPV), and symptomatic treatment (ST). In-hospital mortality at 30 days was the primary outcome.

Results: From 18,184 patients admitted to the hospital, a total of 3,193 moderate to severe COVID-19 cases were included. Of these, 2,256 (70.65%) received FPV with dexa, 828 (25.93%) received FPV, and 109 (3.41%) received ST. The restricted mean survival times for FPV with dexa was 29.68 days (95% CI: 29.52, 29.84), FPV was 29.46 days (95% CI: 29.22, 29.71), and ST was 28.14 days (95% CI: 26.51, 29.76). Only FPV showed marginal significant difference when compared to ST. However, there was a trend in prolonging survival time in FPV with dexa and the results were more pronounced in patients with severe and hypoxic groups.

Conclusion: Our emulated target trial suggests favipiravir, especially with dexamethasone, offers modest survival benefit in moderate to severe COVID-19, particularly in hypoxic patients. It supports favipiravir as a practical antiviral in setting where other antivirals are not available. Further controlled studies are needed to confirm its role alongside standard corticosteroid therapy.

Keywords: Favipiravir, COVID-19, In-hospital mortality, Target trial emulation,

2- Introduction

Strengths:

- Provides a detailed background on the role of favipiravir and its widespread use during the pandemic.

- Highlights the gap in evidence for its efficacy in moderate to severe COVID-19 cases, setting a strong rationale for the study.

Criticisms:

- Redundancy in describing the historical context of favipiravir’s use and the delta wave pandemic. While informative, some details could be streamlined.

- Limited critical analysis of conflicting evidence regarding favipiravir’s efficacy from past studies.

Suggestions for Improvement:

- Condense the description of favipiravir’s history to focus on its relevance to Thailand’s healthcare system.

- Introduce a clearer comparison of existing studies, highlighting where this study fills a knowledge gap.

Response

Thank you very much for the suggestion. We have revised our introduction as follows:

“Coronavirus disease 2019 (COVID-19) is a highly transmissible, severe acute respiratory illness caused by the coronavirus SARS-CoV- presenting a significant challenge to global healthcare systems and prompting extensive investigations into several antiviral agents [1, 2]. In Thailand, as in other countries, favipiravir was rapidly adopted as a first-line antiviral treatment for COVID-19 following its emergency authorization in 2020. [3-6] . This drug was included in the Thai Ministry of Public Health (MoPH) guidelines and widely used during the peak of the pandemic, particularly in response to the third phase that was driven by the delta variant in 2021 [7]. This variant significantly increased the number of patients requiring hospitalization, associated with higher mortality, and transmissibility [8, 9]. The surge in cases led to the establishment of Thailand’s largest national field hospital to manage patient overflow and ease the burden on healthcare facilities.

Despite its widespread use, favipiravir’s clinical efficacy in COVID-19 remains controversial [1, 10]. Several clinical trials focused primarily on mild COVID-19 cases, where favipiravir's effect is minimal, as most patients naturally recover with supportive care [11, 12]. However, in moderate to severe patients who are facing higher risks of respiratory failure and complications, antiviral and anti-inflammatory therapies may play a crucial role in improving outcomes [13, 14]. In addition, the study from RECOVERY trial demonstrated that dexamethasone, a potent anti-inflammatory steroid, reduced 28-day mortality in COVID-19 patients but not among those receiving no respiratory support [15]. Therefore, during the pandemic wave, the Thai MoPH guideline suggested the use of favipiravir in combination dexamethasone in patients with moderate to severe symptom.

In Thailand, studies investigated the use of favipiravir in COVID-19 patients are partly limited. Studies by Sirijatuphat et al. [16] and Siripongboonsitti et al. [17] compare efficacy of favipiravir to control treatment and found that favipiravir showed improvements on clinical outcomes in mild to moderate COVID-10 patients. Rattanaumpawan et al. [18]reported the favipiravir’s effectiveness in reducing hospital stays among hospitalized patients, though its impact on mortality remained unclear. Furthermore, a real-world study during the Delta variant wave indicated that favipiravir monotherapy reduced the 28-day mortality risk in severe COVID-19 (Relative risk = 0.72 (95% CI 0.58–0.91; P = 0.006) but not in mild and moderate patients [19]. The results are still inconclusive due to difference in trial protocols. Additionally, causal effect if favipiravir in combination with dexamethasone in moderate to severe patients remain unexplored.

Therefore, this study aimed to evaluate the effectiveness of favipiravir on in-hospital mortality among patients with moderate to severe symptoms admitted to our field hospital setting. We compared patients who received favipiravir monotherapy or with additional systemic dexamethasone treatment to those who received symptomatic treatment based on standard care. Our main objective was to provide causal evidence of the treatments through an emulation of a target trial from observational data of the largest field hospital in Thailand.”

3- Methods

Strengths:

- The study design is robust, with careful consideration of biases such as immortal time bias.

- Comprehensive description of statistical techniques, including inverse probability of censoring weighting (IPCW) and restricted mean survival time (RMST) analysis.

Criticisms:

- The description of cloning and censoring might confuse readers unfamiliar with advanced epidemiological methods.

- Simplification or clarification is necessary.

- Potential residual confounding due to unmeasured variables (e.g., adherence to treatment protocols) is not addressed.

Suggestions for Improvement:

- Add a brief supplementary explanation or diagram illustrating the cloning-censoring process.

- Acknowledge the potential for residual confounding and discuss its impact on results.

Response

Thank you fir the suggestion in explanation imprivement. We have revised the study flow diagram in Figure 2 to imrove the demonstration of emulate target trial.

Furthermore, we have revised the discussion related to the methodology including the confounding as follows :

To compare survival outcome of treatments, the emulated target trial methodology is particularly relevant to our context because traditional observational analyses are prone to bias. First, imbalanced prognostic characteristics between treatment arms can result in a baseline confounding bias. Second, different treatment initiation times significantly reflect the immortal time bias [20, 21]. Patients who were assigned to received FPV and end up in the treated arm must survive until the treatment initiation. This artificially contributes to an inflated beneficial effect of FPV if both symptomatic treatment and FPV groups were equally defined at time zero and followed up while FPV groups must survive until prescription and the actual follow up begin. By emulating a trial protocol, we specifically defined a time zero of follow-up at admission for all treatment groups and allowed a grace period to receive FPV for one day after admission. Then, we cloned the patients to tripled and randomized them to each arm, thereby balancing the baseline across all study arms. The cloned data were censored, and the survival time analysis were weighted by censoring probability. Even the propensity score matching studies were also observed in favipiravir efficacy research [22-24], our study provides additional bias adjustment through the cloning, and censoring. From our analysis, the baseline BMI, duration from PCR to admission, underlying except DM and hypertension, and vaccination status were well balanced after weighting, while the remaining were later adjusted in the RMST model. Although the emulating a target trial may not fully replace an actual randomized controlled trial, but in a pandemic setting, this approach offers a rigorous way to extract causal insights from real-world data with minimized baseline imbalance and immortal time bias when RCTs are limited or delayed [25].

4- Results

Strengths:

- Results are well-organized with clear presentation of key findings through Kaplan-Meier curves and RMST analyses.

- Comprehensive demographic data provide context for interpreting the findings.

Criticisms:

- Baseline imbalances between treatment groups (e.g., age and vaccination status) may bias the findings, and while addressed statistically, their potential impact is not discussed adequately.

- The exclusion criteria (e.g., patients with delayed treatment initiation) could introduce selection bias, limiting generalizability.

Suggestions for Improvement:

- Discuss the extent to which baseline imbalances, even after weighting, could affect the study’s conclusions.

- Include sensitivity analyses or stratified results to explore the robustness of findings across subgroups (e.g., age or comorbidity status).

Response เติม method 10% weighting

Due to the emulation of target trial, we create clones for each arm before receiving the trial regimens. Therefore, at the baseline, the baseline of these three groups are equal. However, the weighting process helps improve the baseline as demonstrated in S2 table in supplementary files. We, additionally performed the adjustments in the primary outcome for factors that remain imbalance (STD >10%) after weighting [26]. In our case, after weighting, respiratory supports, gender, age, underlying diseases, and oxygen saturation remained imbalance (S1 Table).

We have explained in statistical analysis section as follows :

We used the standardized difference (STD) to assess the degree of differences in patient characteristics across the three groups (three comparison pairs). A significant difference bet

---

## [Editor Report · Decision Letter 1]

4 May 2025

In-hospital mortality outcome of favipiravir in patients with moderate to severe COVID-19 infection: an emulated target trial using real-world data from the largest field hospital in Thailand

PONE-D-24-51078R1

Dear Dr. Phichayut Phinyo,

We’re pleased to inform you that your manuscript has been judged scientifically suitable for publication and will be formally accepted for publication once it meets all outstanding technical requirements.

Kind regards,

Dr. Chiara Bertagnin

Academic Editor

PLOS ONE

---

## [Editor Report · Acceptance letter]

PONE-D-24-51078R1

PLOS ONE

Dear Dr. Phinyo,

I'm pleased to inform you that your manuscript has been deemed suitable for publication in PLOS ONE. Congratulations! Your manuscript is now being handed over to our production team.

Kind regards,

on behalf of

Dr. Chiara Bertagnin

Academic Editor

PLOS ONE